# Enumerating Circulating Tumor Cells with a Self-Assembled Cell Array (SACA) Chip: A Feasibility Study in Patients with Colorectal Cancer

**DOI:** 10.3390/cancers11010056

**Published:** 2019-01-08

**Authors:** Hsueh-Yao Chu, Long-Sheng Lu, Wanying Cho, Shin-Yao Wu, Yu-Cheng Chang, Chien-Ping Lin, Chih-Yung Yang, Chi-Hung Lin, Jeng-Kai Jiang, Fan-Gang Tseng

**Affiliations:** 1Department of Engineering and System Science, National Tsing Hua University, Hsinchu 30013, Taiwan; kecokoyo@gmail.com (H.-Y.C.); zither0709@gmail.com (W.C.); hedonistalbertwu@gmail.com (S.-Y.W.); jerry3910@yahoo.com.tw (Y.-C.C.); 2Department of Radiation Oncology, Taipei Medical University Hospital, Taipei 11031, Taiwan; lslu@tmu.edu.tw; 3Graduate Institute of Biomedical Materials and Tissue Engineering, College of Biomedical Engineering, Taipei Medical University, Taipei 11031, Taiwan; 4International Ph.D. Program in Biomedical Engineering, College of Biomedical Engineering, Taipei Medical University, Taipei 11031, Taiwan; 5Institute of Nano Engineering and Micro Systems, National Tsing Hua University, Hsinchu 30013, Taiwan; 6Institute of Microbiology and Immunology, National Yang-Ming University, Taipei 11221, Taiwan; jainping@gmail.com (C.-P.L.); linch@ym.edu.tw (C.-H.L.); 7Department Education Research, Taipei City Hospital, Taipei 10341, Taiwan; yc3636@hotmail.com; 8Center for General Education, National United University, Miaoli 36003, Taiwan; 9Ph.D. Program in Pharmaceutical Biotechnology, Fu Jen Catholic University, New Taipei City 24205, Taiwan; 10Division of Colorectal Surgery, Department of Surgery, Veterans General Hospital-Taipei, Taipei 11217, Taiwan; 11Department of Engineering and System Science, Frontier Research Center on Fundamental and Applied Sciences of Matters, National Tsing-Hua University, Hsinchu 30013, Taiwan; 12Research Center for Applied Sciences, Academia Sinica, No. 128, Sec. 2, Academia Rd., Nankang, Taipei 11529, Taiwan

**Keywords:** colorectal cancer (CRC), circulating tumor cell (CTC), liquid biopsy, cancer metastasis

## Abstract

Colorectal cancer (CRC) is the second most common cause of cancer-related death worldwide. Detecting and enumerating circulating tumor cells (CTCs) in patients with colorectal cancer emerged as an important prognostic tool which provides a direct estimate of metastatic potential. Improving the turnaround time and decreasing sample volume is critical for incorporating this liquid biopsy tool into routine practice. The objective of the current study was to validate the clinical feasibility of a self-assembled cell array (SACA) chip, a CTC counting platform with less than 4 h turnaround time, in patients with newly diagnosed colorectal cancers. In total, 179 patients with newly diagnosed colorectal cancers from a single institute were enrolled. Epithelial cell adhesion molecule positive (EpCAM(+)), cluster of differentiation 45 negative (CD45(−)) cells were isolated and enumerated from 2 mL of peripheral vein blood (PB) and inferior mesenteric vein blood (IMV) samples obtained during surgery. We found that the CTC count in PB but not IMV correlates with disease stages. Neoadjuvant chemotherapy did not lead to decreased CTC count in both types of blood samples. With cutoffs of four CTCs per 2 mL of blood, and serum carcinoembryonic antigen (CEA) level of 5 ng/mL, patients with non-metastatic disease were more likely to experience recurrence if they had high PB CTC count and high serum CEA concentration (odds ratio, 8.9). Our study demonstrates the feasibility of enumerating CTCs with a SACA chip in patients with colorectal cancer.

## 1. Introduction

Metastasis is the leading cause of cancer morbidity and mortality [1]. Circulating tumor cells (CTCs) are cells detached from an established tumor which enter the systemic circulation. A fraction of CTCs contain cancer-initiating cells that are capable of self-sustained survival and proliferation for establishing new foci of micrometastases in distal organs, and are thought to be an important mechanistic link for cancer metastasis [2]. Indeed, elevated CTC count is associated with shorter progression-free survival and overall survival in patients with metastatic breast cancer [3,4]. CTC is also detected in the blood of patients with various solid tumors [5,6], and its numbers correlate to clinical stages, disease recurrence, tumor metastasis, treatment outcomes, and prognostic significance [7,8,9,10]. The non-invasive and universally informative nature of CTC analysis heralds the age of “liquid biopsy” research [11,12], which allows serial and real-time monitoring of cancer status to support personalized treatment decisions. 

Colorectal cancer (CRC) is one of the leading causes of death in the world [13,14,15]. It is the third most common malignant tumor and the second leading cause of cancer-related death [16]. The global burden of CRC is expected to increase by 60% in 2030, with more than 2.2 million new cases and 1.1 million cancer deaths [17]. Despite the advances in screening procedures and adjuvant treatment, about 50% of CRC patients will eventually develop metastatic disease. The most effective approach to reducing CRC-related death is early diagnosis of the disease, as well as recurrent events [18,19]. If found early, surgical resection of limited liver and lung metastases can lead to a five-year survival rate of 25 to 45%, which is a great improvement compared to the five-year survival rate of less than 5% in the general population with metastatic CRC. Therefore, there is an emergent need to develop a non-invasive method that allows sensitive and specific diagnosis of subclinical CRC progression to enable timely radical intervention.

CTC enumeration is one of the possible solutions for early detection of primary or recurrent CRC, since CTC can be isolated from the blood of patients with colorectal cancer [20]. However, CTC enumeration is technically challenging. It is well known that CTC is very rare (1–100 in 10^9^ blood cells) [21,22], and numerous approaches are used to detect CTC in clinical samples [23,24,25,26,27]. The most popular approach relies on antibody-based positive selection of cells expressing epithelial cell adhesion molecules (EpCAM) on their surface, followed by immunocytochemical identification of cytokeratin-positive, cluster of differentiation 45 (CD45)-negative nucleated cells [28,29]. Meanwhile, physical CTC enrichment is especially useful when minimal manipulation is desired to allow functional study or cell culture. We recently developed a microfluidic technology that utilizes the gradient strain of directed liquid flow to guide self-assembly of cells into a tightly packed monolayer in less than 15 min. Self-Assembled Cell Array (SACA) Chip, or self-assembled cell array, is capable of performing multiple rounds of high-efficiency fluid exchange on top of a cellular monolayer with minimal cell loss [27]. When coupled to an optimized three-dimensional microwell dialysis (3D-microDialysis) chip [30], we could perform multiplex antibody labeling of CTC for image analysis in less than 4 h. The approach was sensitive to the identification of one CTC out of 10^5^ cells [27]. In this study, we designed an observational clinical research to evaluate the feasibility of enumerating CTC with SACA from whole-blood samples of patients with CRC.

## 2. Results

### 2.1. Fluorescent Micrograph-Based CTC Analysis

#### Process Development in Spike-In Samples

In order to detect rare cells in a small amount of biological fluid samples, SACA chips are designed for a seamless microfluidic process to enrich and immunolabel CTC from whole blood. The SACA chip used in this experiment can detect one target cell spiked into 10,000,000 white blood cells (WBCs; Appendix A). In the current study, patients with CRC were enrolled. We validated a SACA chip with a CRC cell line, namely HCT116. When spike-in HCT116 cells were prepared in leucocyte at 1:1,000,000 dilution, cancer cells could be identified with fluorescein isothiocyanate (FITC)-conjugated anti-EpCAM antibody. Representative micrographs are shown in Figure 1.

### 2.2. Validation in Clinical Samples

#### 2.2.1. Demographics

In total, 132 patients were prospectively enrolled in the study between May 2015 and August 2018 (Table 1). All patients had peripheral vein blood (PB) samples. However, PB from three patients was not analyzable due to technical reasons. Inferior mesenteric vein (IMV) blood samples were available in a subgroup of 95 patients, as some patients with early0stage diseases were managed with conservative surgery and, therefore, the inferior mesenteric vein was not approached during surgery. The cohort consisted of patients with relatively early diseases, as 72% of cases had T2 or lower tumors and 73% of cases were negative for lymph node involvement. However, 17 cases were found with distal metastases in the initial presentation. These cases were treated separately in the further analyses as the disease courses were very different from the rest of the cohort.

#### 2.2.2. CTC and Clinical Relevance

Two milliliters of PB with or without IMV samples were collected from each patient included in this study. CTC detection and enumeration were carried out through the SACA chip system (Table 2). CTCs were identified in 80% of patients (PB: 56/66; IMV: 52/58).

#### 2.2.3. CTCs in PB and IMV of Patients with CRC

With the system established, we were able to identify CTCs from 132 prospectively collected clinical samples. Buffy coat from 2 mL of whole blood was loaded onto the SACA chip and was allowed to form a tightly packed monolayer. The whole preparation was stained and imaged for quantification of CTCs. Representative images from five cases are shown in Figure 2. Both white blood cells and CTCs showed the morphology of live cells. CTCs were identified as Hoechst 33258 positive/EpCAM-FITC positive/CD45-PECy-7 negative cells. WBCs were Hoechst 33258 positive/EpCAM-FITC negative/CD45-PECy-7 positive cells. In addition to single cells, CTC clusters were also found in seven PB and three IMV samples (Appendix A).

### 2.3. Clinical Implications of CTC in Colorectal Cancer

#### 2.3.1. CTC Count Correlates with Stages of Non-Metastatic CRC

The CTC counts and their correlation to neoadjuvant chemotherapy and disease stages are analyzed in Figure 3. CTCs were identified in 84.8% of PB samples (56/66) and 89.6% of IMV samples (52/58). CTC counts in PB and IMV did not show significant difference. Sixteen patients received neoadjuvant chemotherapy (NAC) prior to CTC sampling. The CTC counts in these patients were not significantly different from those without prior chemotherapy either in PB or IMV samples. 

In addition, we found that the mean of CTC count in the PB NAC(−) group increased with disease stage (stage 0 to stage III; mean ± standard error of the mean (SEM) 4.0 ±1.7 to 6.3 ± 1.8, Figure 3A), while the same trend was not observed in the IMV NAC(−) group (Figure 3B). For PB NAC(+) and IMV NAC(+) groups, CTC count did not correlate with disease stages. We also noticed that serum CEA level had a stronger association to disease stages compared to that found for the CTC count (Figure 3C).

#### 2.3.2. CTC Count and CEA Synergize to Predict Recurrence in Patients with Non-Metastatic CRC

In order to test whether CTC counts and serum tumor marker CEA may be prognostic for disease recurrence, we analyzed the odds ratio (OR) for these biomarkers. Disease recurrence was defined as local recurrence, distal metastasis, or death at the time of last clinical contact. For CEA, the cutoff value was set to 5 ng/mL based on the clinical laboratory reference. For PB CTC, the cutoff value was set to four cells per 2 mL of blood, which was the median in this cohort. In non-metastatic patients (stage 0–III), the recurrence rate was 10% for patients with CTC > 4, in contrast to a 3% recurrence rate in patients with CTC ≤ 4 (*p* = 0.00081, OR = 4.00, Figure 4A). The recurrence rate was 11% for patients with serum CEA concentration > 5 ng/mL, in contrast to 4% recurrence rate in patients with CEA ≤ 5 ng/mL (*p* = 0.00046, OR = 2.68, Figure 4B). Interestingly, we found that combination of these two markers better predicts the recurrent event. The recurrence rate was 25% for patients with CEA > 5 ng/mL and CTC > 4, while the recurrence rate was 4% for all the other patients (*p* = 0.00085, OR = 7.16, Figure 4C). We also analyzed these biomarkers in patients with metastatic diseases. In these patients (*n* = 19), the recurrence rate was 50% for patients with CTC > 4, in contrast to 0% recurrence rate for CTC ≤ 4 (*p* = 0.105, Figure 4D). When stratified with CEA, the recurrence rate was 20% for patients with CEA > 5 ng/mL and 0% for patients with CEA ≤ 5 ng/mL (*p* = 0.248, Figure 4E). Combining CEA and CTC, the recurrence rate was 50% for patients with both CEA > 5 ng/mL and CTC > 4, in contrast to 18% for all the other patients (*p* = 0.312, OR = 4.66, Figure 4F). A numerical summary of case numbers and ORs is listed in Table 3.

#### 2.3.3. Survival Analysis 

Kaplan–Meier survival curves on progression-free survival (PFS) in non-metastatic patients were plotted. Subjects without these events at the time of their last visit were censored on that day. Cutoffs were set to four CTCs/2 mL of blood and 5 ng/mL CEA as described in the previous section. There was no statistical significance between PFS of patients with CTC > 4 and in patients with CTC ≤ 4 (Figure 5A). Stratification with CEA did not have statistical significance either (Figure 5B). However, when combining both biomarkers, patients with both CTC > 4 and CEA > 5 ng/mL had a significantly worse PFS compared to others that did not meet the criteria (Figure 5C).

## 3. Discussion

In this study, we showed that the SACA chip could be reliably used to enrich rare cancer cells (Figure 1). This device was able to identify CTCs and CTC clusters in clinical samples based on fluorescence-conjugated antibodies (Figure 2 and Appendix A). We also compared CTC counts in both PB and IMV in 132 CRC patients. Previous studies [22] showed that the CTC counts were significantly correlated with disease stage. Here, we showed a stage-related increase of PB CTC counts in patients with non-metastatic diseases (Figure 3A), and this trend did not exist between IMV CTC counts and CRC disease stages. Moreover, we did not observe a significantly higher count of CTC in IMV compared to PB. It is predicted that IMV CTC count will be higher in advanced disease stages, and it will be higher than PB CTC count because of first-pass effects secondary to the drainage pattern of colon cancers. It remains unclear why IMV CTC counts in our cohort did not support the prediction; therefore, more research will be necessary to define the clinical implication of IMV CTC count.

The serum CEA level (Figure 3C) showed a positive correlation with disease stages in the PB of patients, just like PB CTC count. Since CEA and CTC measure different biological processes, we speculated that both biomarkers may work in concert to better predict clinical outcome of CRC. Indeed, our data support this idea. More recurrent events and shorter progression-free survival in non-metastatic CRC patients could be best predicted by defining a high-risk group with high CEA and PB CTC counts at baseline. An important clinical implication is that escalating adjuvant treatment for this subgroup of patients may improve overall survival outcome in CRC patients. Moreover, elevated CEA and CTC count may be used serially along the disease course for early identification of disease recurrence before actionable early metastases are big enough to be detected by routine imaging modalities. This may facilitate the timely delivery of systemic therapy to destroy favorable tumor microenvironments and avoid formation of disseminated subclinical lesions, which may be an effective approach to improve patients’ overall survival.

## 4. Materials and Methods 

### 4.1. Clinical Sample Acquisition

The prospective trial was conducted at the General Clinical Center for colorectal surgery in Taipei, Taiwan. The objective was to evaluate the consistency of CTC number and to analyze the corresponding changes of different clinical stages via images and CTC counts. From May 2015 to August 2018, 179 subjects were enrolled with a median follow-up of 159 days. The protocol was approved by the Institutional Review Board of Taipei Veterans General Hospital and the research was registered in clinical trials (IRB-TVPEGH: 2016-02-008CC; 2017-06-018BC). All patients agreed and signed their informed consent.

### 4.2. Sample Preparation Process for Clinical Research

#### 4.2.1. Blood Collection and CTC Isolation 

PB was collected before the treatment began (baseline) and IMV was collected during surgery in order to conduct CTC evaluation. Samples were kept in plastic blood collection tubes with K2EDTA (BD Vacutainer^®^, Plymouth, UK) at room temperature and were processed within 24 h of collection. The isolation procedure began with the Leucosep TM tube (bio-check LABORATORIES LTD, New Taipei City, Taiwan) to remove red blood cells from whole blood. In this step, red blood cells were removed through Ficoll–Paque PLUS (GE Healthcare Life Sciences, Taipei, Taiwan) by concentration gradient. The mononucleated cells containing CTC (peripheral blood mononuclear cells (PBMCs)) after centrifugation were collected, and the CTC cell retention rate was about 75–85% [30].

#### 4.2.2. Immunofluorescence Staining

The PBMC pellet immunofluorescence assays were labeled to detect CTC with Hoechst33258 (Thermo Fisher Scientific Taiwan Co., Ltd, Taipei, Taiwan) staining at 37 °C for 30 min, and then EpCAM-FITC (BioMab Inc., Taipei, Taiwan) and CD45-pecy7 (Beckman Coulter Inc, Brea, CA, USA) antibody staining at 37 °C for 30 min. Hoechst staining is part of the family of blue fluorescent dyes used to stain cell nucleus DNA. CD45 is a 180–240-kDa glycoprotein, also known as leukocyte common antigen (LCA), which recognizes white blood cells in PBMC, and almost all cancers are expressed as EpCAM antigens; thus, EpCAM epithelial differentiation antibodies were used to identify CTCs, while FITC is a fluorescent dye with a molecular weight of 389 kDa. The SACA chip system was used in CTC enumeration the repeatability, reproducibility, accuracy, and technical details of the SACA chip were described previously [27].

#### 4.2.3. Imaging

Fluorescence imaging was carried out using a normal vertical optical microscope and fluorescence microscope ((Olympus IX71) Olympus Taiwan Co., Ltd., Taichung, Taiwan). The exposure time for each sample was kept at 0.1 s for each appropriate wavelength for different dyes.

#### 4.2.4. The Preparation Time of Each Step of the Clinical Test Sample

Figure 6 shows the experiment process flow of all clinical samples. It took about 25 min to separate the PBMC from the blood, and about 60 min to dye the cell. The SACA chip system was used in CTC enumeration, and the cells were self-assembled on the SACA chip for 5 min. Finally, the images were observed using a fluorescence microscope for about 3 h. The entire sample processing took about 4.5 h.

### 4.3. Cell Line Culture

The HCT116 cell line (American Type Culture Collection (ATCC^®^, CCL-2™, Building Construction Resource Center, Inc., Hsinchu, Taiwan)) involves human colon cancer cells, and they were injected into the white blood cells in order to simulate CTC in patients’ blood. HCT116 cell lines was cultured in a medium containing Roswell Park Memorial Institute (RPMI-1640) (Corning INC., Brooklyn, NY, USA), 10% fetal bovine serum (FBS) (Corning INC., Brooklyn, NY, USA), 1% penicillin/streptomycin (Pen-Strep) (GIBCO^®^, New York, NY, USA) at 37 °C and 95% air and 5% CO_2_.

### 4.4. Cell Line Spike-In Controls

In order to evaluate the feasibility of the CTC enrichment process of in this study, HCT116 cells were cultured in a flask at 37 °C, 95% air, and 5% carbon dioxide for three days, and 0.05% trypsin/ethylenediaminetetraacetic acid (EDTA) (GIBCO) was used to dissociate the link between the monolayer and the culture dish before spiking-in whole-blood samples.

### 4.5. Statistical Analysis

The values are expressed as means ± SEM. The significance level was set at *p* < 0.05 (both sides). All statistical analyses were performed using SPSS version 12.0 (IBM, Taipei, Taiwan) and GraphPad Prism 6 (GraphPad Software Inc. San Diego, CA, USA) for Windows. Progression-free survival (PFS) referred to the time between the day of blood sampling and the day of either local recurrence or distal metastasis or death. Subjects without these events at the time of their last visit were censored on that day.

## 5. Conclusions

Metastasis is a major cause of death from various solid tumors, and it is often associated with poor prognosis. CTC isolation and characterization emerged as an exciting field of study and provided us with unprecedented opportunities to detect early micrometastases for radical interventions. Our study successfully demonstrates the feasibility of SACA, a novel CTC enumerating platform, to obtain clinically meaningful information for CRC patients. We also provide evidence that combining CTC count from SACA and CEA levels is a strong risk stratification tool. Since SACA is a quick platform for reliable CTC detection, its potential to assist in clinical liquid biopsy in all spectra of CRC management warrants further investigation.

## Figures and Tables

**Figure 1 cancers-11-00056-f001:**
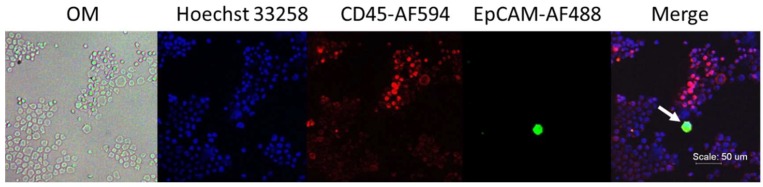
Phenotypic analysis in spike-in samples. The HCT116–leucocyte suspension at 1:1,000,000 dilution was prepared into a monolayer with a self-assembled cell array (SACA) chip and was stained with anti-epithelial cell adhesion molecule (EpCAM) conjugated with fluorescein isothiocyanate (FITC), anti-cluster of differentiation 45 (CD45)-AF594, and Hoechst 33258. The target cell is marked with a white arrow.

**Figure 2 cancers-11-00056-f002:**
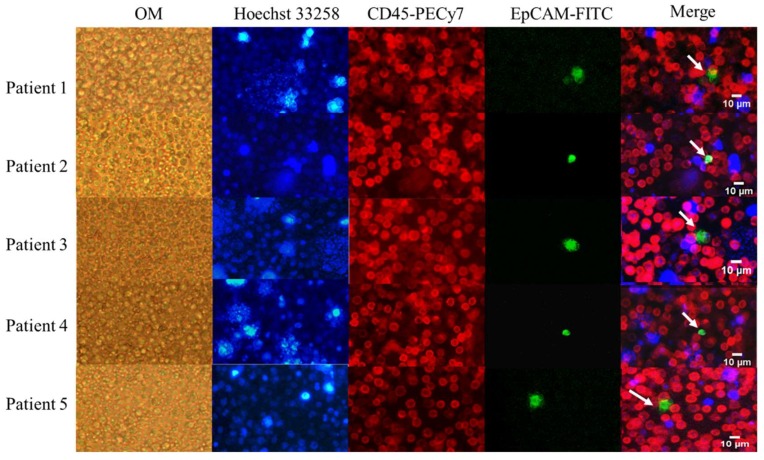
Detection of circulating tumor cells (CTCs) in peripheral vein blood (PB) from colorectal cancer (CRC) patients. The images were acquired on the microscope at 10× magnification. OM: bright-field images under optical microscopy. The target cells are marked with white arrows. Blue: Hoechst 33258; Red: CD45-PECy-7; Green: EpCAM-FITC.

**Figure 3 cancers-11-00056-f003:**
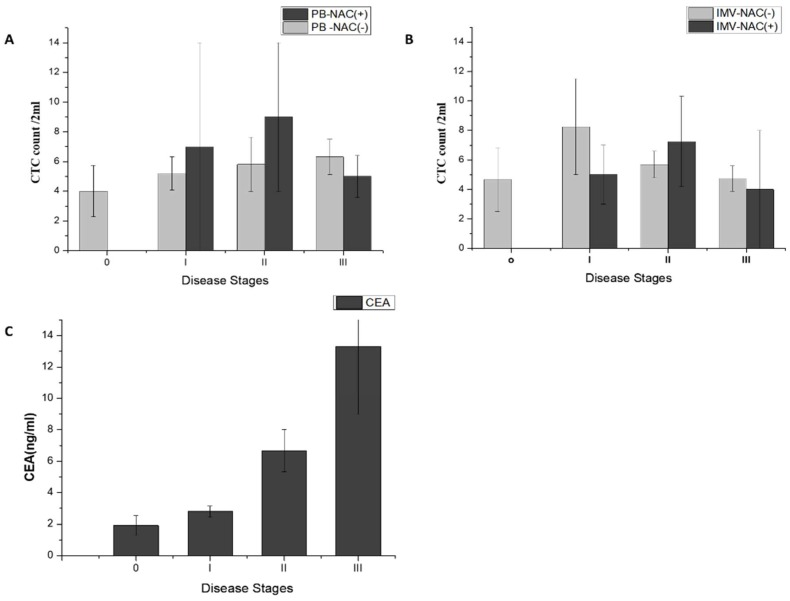
Analysis of the relationships between disease stage and blood biomarkers, stratified by the presence or absence of neoadjuvant chemotherapy (NAC). (**A**) Peripheral vein blood (PB) and clinical staging analysis of NAC(−) and NAC(+) patients; (**B**) Inferior mesenteric vein blood( IMV) and clinical staging analysis of NAC(−)and NAC(+) patients; (**C**) the carcinoembryonic antigen( CEA) concentrations and disease stages.

**Figure 4 cancers-11-00056-f004:**
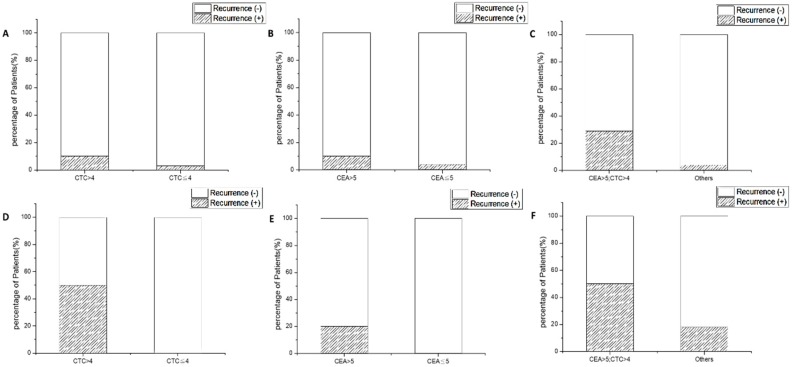
The recurrence rates in patients stratified by their CTC and carcinoembryonic antigen (CEA) counts, and their combination. (**A**) Recurrence rates of different CTC levels in non-metastatic patients; (**B**) recurrence rates of different CEA levels in non-metastatic patients; (**C**) recurrence rates of different CTC and CEA levels in non-metastatic patients; (**D**) recurrence rates of different CTC levels in metastatic patients; (**E**) recurrence rates of different CEA levels in metastatic patients; (**F**) recurrence rates of different CTC and CEA levels in metastatic patients.

**Figure 5 cancers-11-00056-f005:**
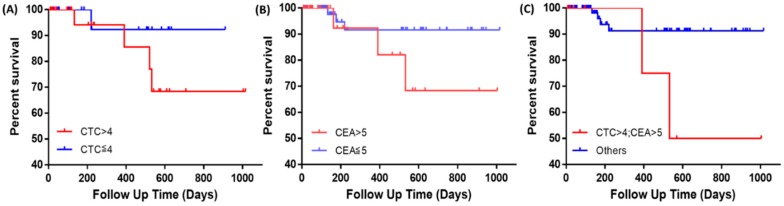
Kaplan–Meier survival curves of progression-free survival in patients with non-metastatic diseases, stratified with CTC and CEA. (**A**) Red line: CTC > 4; blue line: CTC ≤ 4; *p* = 0.242; (**B**) red line: CEA > 5 ng/mL; blue line: CEA ≤ 5 ng/mL, *p* = 0.176; (**C**) red line: both CTC > 4 and CEA > 5 ng/mL; blue line: others, *p* = 0.033.

**Figure 6 cancers-11-00056-f006:**
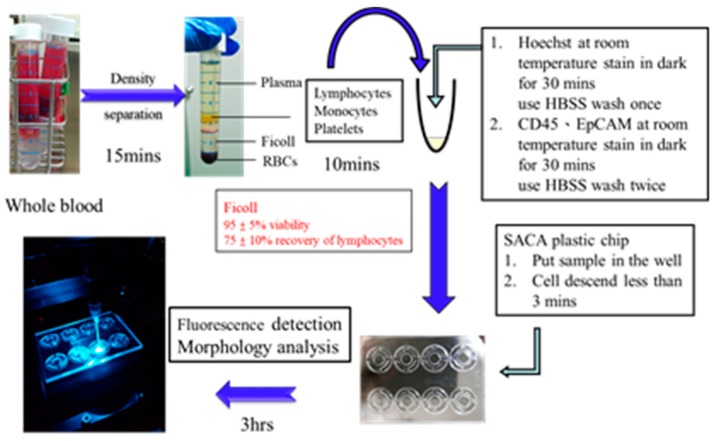
Sample preparation process for clinical trial.

**Table 1 cancers-11-00056-t001:** Demographics of patients. Age, sex, tumor location, and TNM (Tumor, Nodes, Metastasis-classification) stages were separately analyzed for patients with peripheral blood (PB) (**A**) or inferior mesenteric vein blood (IMV) (**B**) samples. Lymph node involvement, carcinoembryonic antigen (CEA), and carbohydrate antigen 19-9 (CA19-9) levels in the whole cohort were also analyzed (**C**).

**(A)** **Peripheral Vein Blood (PB)**		
Groups	Patients (*n*)	Percentage (%)
Age (years)		
<60	55	43%
≥60	74	57%
Sex		
Male	75	58%
Female	54	42%
Tumor location of the disease		
Colon	89	69%
Others	40	31%
T classification (TNM Stage)		
Earlier than T2	123	72%
Later than T3	48	28%
**(B)** **Inferior Mesenteric Vein Blood (IMV)**		
Groups	Patients (*n*)	Percentage (%)
Age (years)		
<60	47	49%
≥60	48	51%
Sex		
Male	56	59%
Female	39	41%
Tumor location of the disease		
Colon	64	67%
Others	31	33%
T classification (TNM Stage)		
Earlier thanT2	60	63%
Later than T3	35	37%
**(C)** **Lymph Node Metastasis**		
Groups	Patients (*n*)	Percentage (%)
Negative	82	73%
Positive	30	27%
Preoperative Serum CEA (ng/mL)		
	Patients (*n*)	Percentage (%)
≤5	81	65%
>5	44	35%
Preoperative Serum CA19-9 (U/mL)		
	Patients (*n*)	Percentage (%)
<35	106	87%
≥35	16	13%

TNM: Tumor, Nodes, Metastasis-classification

**Table 2 cancers-11-00056-t002:** Number of circulating tumor cells (CTCs) detected in the colorectal cancer cases.

**(A)** **CTC Count (PB)—2 mL of Blood**				
Disease stages	No. of cases	Mean ± SEM	Range (median)	Mode
Non-colorectal cancer cases, NAC (−)				
Benign	7	3.28 ± 0.7	0–6 (3)	3
Non-colorectal cancer cases, NAC (+)				
Benign	1	1 ± N/A	N/A	N/A
Colorectal cancer cases, NAC (−)				
Stage 0	3	4 ± 1.7	1–7 (4)	N/A
Stage I	29	5.62 ± 1.12	1–26 (4)	4
Stage II	34	5.8 ± 1.83	0–57 (4)	4
Stage III	29	6.3 ± 1.18	0–30 (3.5)	4
Stage IV	10	4.6 ± 1.04	1–12 (3)	3
Colorectal cancer cases, NAC (+)				
Stage 0	0	N/A	N/A	N/A
Stage I	2	7 ± 7.0	0–14 (7)	N/A
Stage II	5	9 ± 5.4	1–30 (3)	N/A
Stage III	2	5 ± 1.0	4–6 (5)	N/A
Stage IV	7	2 ± 0.48	1–5 (2)	2
**(B)** **CTC Count (IMV)—2 mL of Blood**				
Disease stages	No. of cases	Mean ± SEM	Range (median)	Mode
Non-colorectal cancer cases, NAC (−)				
Benign	4	6.5 ± 1.50	3–9 (5)	3
Non-colorectal cancer cases, NAC(+)				
Benign	1	0 ± N/A	N/A	N/A
Colorectal cancer cases, NAC (−)				
Stage 0	3	4.6 ± 2.18	2–9 (3)	N/A
Stage I	20	8.25 ± 3.26	0–69 (5)	4
Stage II	27	5.6 ± 0.91	0–21 (6)	7
Stage III	19	4.7 ± 0.86	0–14 (3)	3
Stage IV	7	6.8 ± 1.14	4–13 (6)	5
Colorectal cancer cases, NAC (+)				
Stage 0	0	N/A	N/A	N/A
Stage I	2	7 ± 2.0	0–14 (7)	N/A
Stage II	4	7.2 ± 3.06	1–13 (7)	N/A
Stage III	2	4 ± 4.0	0–8 (4)	N/A
Stage IV	7	4.7 ± 0.83	1–7 (5)	7

NAC—neoadjuvant chemotherapy; SEM—standard error of the mean; N/A—not available.

**Table 3 cancers-11-00056-t003:** Odds ratios (ORs) of CTC, CEA, and their combination to predict recurrent events. Data from patients with non-metastatic CRC are listed in (A), and data from patients with metastatic CRC are listed in (B). N/A: not analyzable due to absence of recurrent event.

**(A)**	**No. of Cases**		**Probability Ratio**
Disease stages 0–III	Recurrence (+)	Recurrence (−)	(odds ratio, OR)
CTC > 4	3	27	4.00
CTC ≤ 4	1	36	
CEA > 5	3	25	2.68
CEA ≤ 5	3	67	
CEA > 5; CTC > 4	2	6	7.16
Others	3	86	
**(B)**	**No. of Cases**		**Probability Ratio**
Disease stage IV	Recurrence (+)	Recurrence (−)	(odds ratio, OR)
CTC > 4	1	1	N/A
CTC ≤ 4	0	7	
CEA > 5	3	12	N/A
CEA ≤ 5	0	3	
CEA > 5; CTC > 4	1	1	4.66
Others	3	14	

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
