# Peer review of "Enumerating Circulating Tumor Cells with a Self-Assembled Cell Array (SACA) Chip: A Feasibility Study in Patients with Colorectal Cancer"

_cancers, 2019, doi:10.3390/cancers11010056_

Round 1

Reviewer 1 Report

General comment

The paper of Hsueh-Yao Chu and coll., report data of an observational prospective study aimed to demonstrate that SACA chip is a feasible tool for enumerating CTCs in colorectal cancer patients.

From what a reader can understand, SACA chip belong to the category of “physical” procedures for collecting CTCs that then may be enumerated. In principle, the paper contains novelty and should be shared with the scientific community, but not in the present form.

Specific comments

Line 34: CTCs are rare events, it raises doubt that reducing the volume of PB examined, may improve sensibility of any CTC assay. The authors should comment this point.

Line 49 and following: The introduction should be extensively revised. The authors are using inappropriate references (e.g. 1, 2 and 8; 25-28) to describe the state of art on CTC detection that seems approximate and superficial. If they focus on the “physical” procedures of CTC enrichment, presenting limits of concurrent methodologies and the plus of their own, the paper will gain soundness.

Line 93: Explain, please, why IMV samples were available only for 95 patients.

Line 106: Which cell line has been used for spiking-samples? In the main test, the authors mention BT474, despite in the supplementary data the photo gallery show Hela results. By the way, Hela cell line do not seem the better choice, considering that is an adherent cell line.

Line 110: What indicate the white ring in the photo? Please explain in the Figure’ legend.

The line’ number has been lost in the following part of the manuscript….

Figure 2: In the merge photos of patients no. 3 and 5, the green signal of FITC is invisible….

The line’ number restart in the following part of the manuscript….

Line 118 and following: Data are commented in the same manner, despite in some analyses no statistically significance difference was found. The paragraph should be revised and commented, with the help of an expert statistician.

Line 150: The manuscript do not show images of CTC clusters, only a plot, about this, in supplementary file…. Insert a photo, please, or erase the comment.

Line 154: “a trend of positive correlation” is a discursive description, of none statistic meaning.

Line 165: “independent predictor” seems to be too stronger definition, since none Kaplan-Meier survival curve is shown in the manuscript.

Supplementary Materials: The description of SACA is totally incomprehensible for a CTC researcher, sorry, not expert in fluid physic. Consider to revise it, please. Clearness is mandatory for novelty, especially if is invading a new research area and increase soundness of your data.

Author Response

We appreciate the comments from the reviewers, which significantly help us improve this manuscript. We provide a point-to-point response in the followings.

Point 1: CTCs are rare events, it raises doubt that reducing the volume of PB examined, may improve sensibility of any CTC assay. The authors should comment this point.

Response 1: We have pointed out in the revised manuscript that SACA chips are designed for a seamless microfluidic CTC isolation and enrichment from whole blood samples. It is a physical device with very high yield rate. Therefore, CTC can be effectively identified in relatively small volume of blood. The relevant description has been updated in LINES 111-114.

Point 2: Line 49 and following: The introduction should be extensively revised. The authors are using inappropriate references (e.g. 1, 2 and 8; 25-28) to describe the state of art on CTC detection that seems approximate and superficial. If they focus on the “physical” procedures of CTC enrichment, presenting limits of concurrent methodologies and the plus of their own, the paper will gain soundness.

Response 2: We have updated the literature cited. Also we pointed out the physical nature of this device in LINES 55-79

Point 3: Line 93: Explain, please, why IMV samples were available only for 95 patients.

Response 3: Some patients with early stage of disease were managed with conservative surgery and therefore inferior mesenteric vein was not approached during surgery.

Point 4: Which cell line has been used for spiking-samples? In the main test, the authors mention BT474, despite in the supplementary data the photo gallery show Hela results. By the way, Hela cell line do not seem the better choice, considering that is an adherent cell line.

Response 4: We changed the target cell line and used HCT 116 colorectal cancer cells for spike-in experiment (Figure 1, line116-117). The HeLa experiment was done to validate the performance of rare cell detection of SACA chip.

Point 5: What indicates the white ring in the photo? Please explain in the Figure’ legend

Response 5: We updated the mark to an white arrow in figure 1 and explained about that in the legend.

Point 6: In the merge photos of patients no. 3 and 5, the green signal of FITC is invisible

Response 6: It has been fixed.

Point 7: Line 118 and following: Data are commented in the same manner, despite in some analyses no statistically significance difference was found. The paragraph should be revised and commented, with the help of an expert statistician.

Response 7: The paragraph has been revised and adequately commented.

Point 8:Line 150 The manuscript do not show images of CTC clusters, only a plot, about this, in supplementary file…. Insert a photo, please, or erase the comment.

Response 8: Pictures and explanations have been added to the supplementary data.

Point 9: Line 154: “a trend of positive correlation” is a discursive description, of none statistic meaning.

Response 9: We have updated the paragraph and avoided the indicated phrase.

Point 10: Line 165: “independent predictor” seems to be too stronger definition, since none Kaplan-Meier survival curve is shown in the manuscript.

Response 10: We did multivariate analysis trying to identify independent predictors for progression free survival. However, only 68 cases were eligible for analysis and we could not find an independent predictor with such a small sample size. Therefore we down-tone our claims in the revised manuscript. In response to the comment, Kaplan-Meier survival curve analysis has been added in Figure 5. Indeed, high CTC and high CEA defines a population with significantly short PFS. However, the software (GraphPad Prism.6) that analyzed the median survival could not calculate the median survival of all data, so this information was not written in Figure 5 analysis information.

Reviewer 2 Report

The feasibility study by Chu et al. analysed enumeration of CTCs using SACA chips, a microfluidic platform for enumarating CTCs and its utility in isolationg CTCs from patients with CRC at various stages. Overall, the study is well executed and performed using a decent number of patients. I would recommend publication once the following are addressed:

1) Why were BT474 cells (a breast carcinoma cell line) used for spike in studies? Although this is an EpCAM positive cell line, since this study looked at CRC patients, could the authors also validate using a CRC line such as HT116 or HT29?

2) In Figure 1, there are numerous regions of CD45-PECy7 and EpCAM-FITC positivity. WHat are these?

3) It would be useful to have a graph showing CTC count/2 ml regardless of NAC treatment per disease stage.

4) Although 4 CTCs were used as the cut-off for statistical analysis, 5 CTCs is usually the cut-off. Can an additional analysis be performed to determine the recurrence in this cohort?

5) Please provide Kaplain-Meier curves to show each event in recurred vs. non-recurred patient. Along with this, please rovide p values and confidence intervals along with the odds ratio.

6) Line 169, please change "prospective trial" to "feasibility study".

7) Please give details of the cell staining procedure.

8) Line 208 and 213. The HeLa cell line is mentioned here but is not used at any point in the study. Please clarify.

9) No details are given for the actual statistical tests used. Please include these in the methods.

10) There are numerous typographical errors throughout the manuscript. Please correct.  

Author Response

Response to Reviewer 2 Comments

We appreciate the comments from the reviewers, which significantly help us improve this manuscript. We provide a point-to-point response in the followings.

Point 1: Why were BT474 cells (a breast carcinoma cell line) used for spike in studies? Although this is an EpCAM positive cell line, since this study looked at CRC patients, could the authors also validate using a CRC line such as HT116 or HT29?

Response 1: Thank you for your comments. We changed the target cell line and used HCT 116 colorectal cancer cells for spike-in experiment (Figure 1)

Point 2: In Figure 1, there are numerous regions of CD45-PECy7 and EpCAM-FITC positivity. What are these?

Response 2: Thank you for your comments. Because FITC’s emission maximum is around 530 nm. However, FITC’s emission is not restricted to green photons; it also emits yellow, orange, red, albeit at lower probabilities. Therefore, it can be seen that CD45 - PECy7 under the excitation of FITC is orange (figure 1.middle), not red (red if activated by PECy7, figure 1.left), and we have replaced this figure (figure 1).

Point 3: It would be useful to have a graph showing CTC count/2 ml regardless of NAC treatment per disease stage.

Response 3: Thank you for your comments. CTC count/2 ml regardless of NAC treatment per disease stage analysis has been added (Table 2).

Point 4: Although 4 CTCs were used as the cut-off for statistical analysis, 5 CTCs is usually the cut-off. Can an additional analysis be performed to determine the recurrence in this cohort? Response 4: Thank you for your comments. CTC cutoff value for CRC is actually 3 / 7.5ml in CellSearch system (Cohen SJ, Punt CJA, Iannotti N, et al. J Clin Oncol. 2008;26(19):3213-3221). In response to your request, we performed additional analysis with CTC cutoff =  5 cells. The results are consistant with the original analysis (figure 4A&4E and Table 3).

Point 5: Please provide Kaplain-Meier curves to show each event in recurred vs. non-recurred patient. Along with this, please rovide p-values and confidence intervals along with the odds ratio.

Response 5: Thank you for your comments. Kaplan-Meier survival curve analysis has been added (Figure 5). (Line160-166)Kaplan-Meier survival curves on progression free survival in non-metastatic patients were plotted. Cutoffs were set to 4 CTC / 2 ml and 5ng/ml of CEA. The results showed that in patients with CTC > 4(red line), in patients with CTC4(blue line),  and there was no statistically significant difference (p = 0.242, Figure 5A). In patients with CEA > 5(red line),; in patients with CEA5(blue line), and there was no statistically significant difference (p = 0.176, Figure 5B); for patients with both CTC>4 and CEA>5, the median survival was 774 days (red line), and for the others (blue line), which is statistically significant (p = 0.033, Figure 5C). However, the software (GraphPad Prism.6) that analyzed the median survival  could not calculate the median survival of all data, so this information was not written in  Figure 5 analysis information.

Point 6: Line 169, please change "prospective trial" to "feasibility study".

Response 6: Thank you for your comments. Has been changed "prospective trial" to "feasibility study".

Point 7: Please give details of the cell staining procedure.

.Response 7: Thank you for your comments. staining procedure has been added (line218-227).

Point 8: Line 208 and 213. The HeLa cell line is mentioned here but is not used at any point in the study. Please clarify.

Response 8: Thank you for your comments. HeLa cell line was used to measure the performance of rare cell enrichment. The data was shown in supplementary Figure S1.

Point 9: No details are given for the actual statistical tests used. Please include these in the methods.

Response9 : Thank you for your comments. Statistical analysis methods have been added in

Point 10: There are numerous typographical errors throughout the manuscript. Please correct.  Response 10: Thank you for your comments. The typographical errors have also been corrected

Round 2

Reviewer 1 Report

General comment

The revised version of the paper is improved. However, before publishing some minor changes need yet.

Specific comments:

Line 75: “(1-100 in 109 blood cells)”…. Did the authors intend “(1-100 in 109 blood cells)”?

Line 91, 92 and 103: In my opinion, the authors should place this paragraph and the following, after the 2.2 Section, Spike-in study paragraph, since this is the standard work-flow: a) development in spike-in samples, b) validation in clinical samples.

Line 95: Insert here, please, the response 3 of the file “Response to Reviewer 1 Comments”.

Line 115: Verify, please, and amend the dilutions of spiked samples. Indeed, there are some inconsistences among dilution presented here (1:1000) and that of Supplementary file: from 1:100 to 1:10.000.000 in the main text, but from 1:10.000 to 1:10.000.000 in the table of Figure S1.

Line 248: The description of Statistical analysis is incomplete. The authors should describe criteria used for PFS, events and censored patients.

Author Response

Response to Reviewer 1 Comments-Round 2

We appreciate the comments from the reviewers, which significantly help us improve this manuscript. We provide a point-to-point response in the followings.

Point 1: Line 75: “(1-100 in 109 blood cells)”…. Did the authors intend “(1-100 in 109 blood cells)”?

Response 1: We indeed mean "(1-100 in 109 blood cells)", It has been fixed in LINES 77.

Point 2: Line 91, 92 and 103: In my opinion, the authors should place this paragraph and the following, after the 2.2 Section, Spike-in study paragraph, since this is the standard work-flow: a) development in spike-in samples, b) validation in clinical samples.

Response 2: we have already placed this paragraph and the following, after the Spike-in study paragraph section.

Point 3: Line 95: Insert here, please, the response 3 of the file “Response to Reviewer 1 Comments”.

Response 3: We've inserted “Inferior mesenteric vein (IMV) blood samples were available in a subgroup of 95 patients, as some patients with early stage diseases were managed with conservative surgery and therefore inferior mesenteric vein was not approached during surgery.  “ in LINES 110-112.

Point 4: Line 115: Verify, please, and amend the dilutions of spiked samples. Indeed, there are some inconsistences among dilution presented here (1:1000) and that of Supplementary file: from 1:100 to 1:10.000.000 in the main text, but from 1:10.000 to 1:10.000.000 in the table of Figure S1.

Response 4: The paragraph has been revised in LINES 100,103-104.

Point 5: Line 248: The description of Statistical analysis is incomplete. The authors should describe criteria used for PFS, events and censored patients.

Response 5: Progression free survival (PFS) referred to the time between the day of blood sampling and the day of either local recurrence or distal metastasis or death. Subjects without these events at the time of their last visit were censored at that day.
